# Effects of Environmental Chemical Pollutants on Microbiome Diversity: Insights from Shotgun Metagenomics

**DOI:** 10.3390/toxics13020142

**Published:** 2025-02-19

**Authors:** Seid Muhie, Aarti Gautam, John Mylroie, Bintu Sowe, Ross Campbell, Edward J. Perkins, Rasha Hammamieh, Natàlia Garcia-Reyero

**Affiliations:** 1Medical Readiness Systems Biology, Center for Military Psychiatry and Neuroscience, Walter Reed Army Institute of Research, Silver Spring, MD 20910, USA; smuhie@genevausa.org (S.M.); aarti.gautam.civ@health.mil (A.G.); bintu.j.sowe.ctr@health.mil (B.S.); rcampbell@genevausa.org (R.C.); 2The Geneva Foundation, Silver Spring, MD 20910, USA; 3U.S. Army Engineer Research and Development Center Environmental Laboratory, Vicksburg, MS 39180, USA; john.e.mylroie@usace.army.mil (J.M.); edward.j.perkins@erdc.dren.mil (E.J.P.); 4Institute for Genomics, Biocomputing & Biotechnology, Mississippi State University, Starkville, MS 39759, USA

**Keywords:** microbiome, metagenomic, chemical pollutants, toxicity, biodiversity

## Abstract

Chemical exposure in the environment can adversely affect the biodiversity of living organisms, particularly when persistent chemicals accumulate over time and disrupt the balance of microbial populations. In this study, we examined how chemical contaminants influence microorganisms in sediment and overlaying water samples collected from the Kinnickinnic, Milwaukee, and Menomonee Rivers near Milwaukee, Wisconsin, USA. We characterized these samples using shotgun metagenomic sequencing to assess microbiome diversity and employed chemical analyses to quantify more than 200 compounds spanning 16 broad classes, including pesticides, industrial products, personal care products, and pharmaceuticals. Integrative and differential comparative analyses of the combined datasets revealed that microbial density, approximated by adjusted total sequence reads, declined with increasing total chemical concentrations. Protozoan, metazoan, and fungal populations were negatively correlated with higher chemical concentrations, whereas certain bacterial (particularly Proteobacteria) and archaeal populations showed positive correlations. As expected, sediment samples exhibited higher concentrations and a wider dynamic range of chemicals compared to water samples. Varying levels of chemical contamination appeared to shape the distribution of microbial taxa, with some bacterial, metazoan, and protozoan populations present only at certain sites or in specific sample types (sediment versus water). These findings suggest that microbial diversity may be linked to both the type and concentration of chemicals present. Additionally, this study demonstrates the potential roles of multiple microbial kingdoms in degrading environmental pollutants, emphasizing the metabolic versatility of bacteria and archaea in processing complex contaminants such as polyaromatic hydrocarbons and bisphenols. Through functional and resistance gene profiling, we observed that multi-kingdom microbial consortia—including bacteria, fungi, and protozoa—can contribute to bioremediation strategies and help restore ecological balance in contaminated ecosystems. This approach may also serve as a valuable proxy for assessing the types and levels of chemical pollutants, as well as their effects on biodiversity.

## 1. Introduction

Microbiomes play an important role in maintaining ecosystem health and functionality. The differential toxicity of various classes of environmental pollutants on these microbial communities has significant implications for ecosystem services, human health, and climate resilience. Particularly, soil and freshwater ecosystems are foundational to agricultural productivity, environmental sanity, and human health but are highly vulnerable to environmental pollutants [1]. Contamination from pharmaceuticals, personal care products, and agricultural runoffs can significantly affect soil and freshwater microbial communities. Chemical pollutants can significantly decrease microbial diversity [2], which can result in a decline in specialized functions, such as nitrogen fixation and organic matter decomposition, which are critical for maintaining soil productivity [3,4,5]. Likewise, pharmaceutical and personal care pollutants reduced the diversity of freshwater microbiomes and altered their functional potential, impairing water quality and ecosystem health [6,7]. In pharmaceutical-contaminated wastewater, the bacterial communities changed dramatically, with the resulting dominant bacterial phylum being Actinobacteria [8]. Long-term antibiotic exposure enriched water microbiomes with general detoxification mechanisms, including those with abundant Quorum Sensing genes [9]. Similarly, long-term industrial chemical contamination shaped/decreased the diversity and activity response of microbiomes at the polluted sites [10]. Changes in microbiome structure and composition in an ecosystem that are caused by chemical contaminates have been shown to have an impact on higher organisms and human health [11].

Assessing the impact of chemical contaminants and the overall scale of risk posed by the usage, disposal, and dispersal of chemicals are important in responding to environmental and health issues [12]. The metagenomic study of microbiomes can be used to assess environmental contamination, including the spatial distribution of pollutants and ecosystem functioning [13]. Understanding the dynamic responses of microbial communities to chemical contamination is essential for forecasting potential long-term changes in ecosystems [14]. Metagenomic profiling of microbiomes can be used as pollution level markers to monitor and develop successful strategies to mitigate the impact of pollutants on ecosystems. For example, microbiomes are used as potential indicators for chemical contaminants, such as for understanding the degradation of aromatic compounds [15]. Furthermore, a meta-taxonomic investigation showed a strong correlation between the abundance of acidophiles and the characteristic physiochemical parameters (metals, acidity, and sulfate) of mining operation discharge, confirming their potential as biomarkers of acid mine drainage pollution [16].

The study presented here focuses on the impact of chemical pollutants on microbiome biodiversity across multiple locations. Concentrations of more than 200 chemical contaminants that belong to pharmaceuticals, personal care products, plasticizers, flame retardants, fuels, polyaromatic hydrocarbons, wastewater indicators (sterols), and other industrial chemicals were measured or estimated in sediment and water samples collected from eight different sites. Then, shotgun metagenomic sequencing and the multiple R package for bioinformatics analytics were used to link the biodiversity of microbiomes to the levels and types of chemical contaminants found in the sediment and water samples. The metagenomic profiling of the sediment and water microbiomes was used (i) to assess if microbiome diversity and membership were affected by chemical contaminants, (ii) to correlate microbial taxa to the identity and levels of chemical pollutants within the sampling ecosystem, and (iii) to infer the adaptation of particular microorganisms, including their implied use for remediation [17]. This comprehensive analysis underscores the importance of metagenomic studies in understanding and mitigating the impact of chemical pollutants on microbial communities and ecosystem health.

## 2. Materials and Methods

Shotgun metagenomic sequencing and chemical analysis were conducted to examine biodiversity and to determine the concentrations of environmental chemical pollutants, respectively, in both water and sediment samples (Appendix A). Processed sequence reads were classified by taxonomy using the NCBI non-redundant nucleotide database. Downstream analyses of the taxonomy, correlations, and plots were carried out using different packages in the R programming language.

### 2.1. Collection of Sediment and Water Samples

Both water and sediment samples were taken from eight different Great Lakes Areas of Concern near Milwaukee, WI, USA, with varying degrees of chemical contaminants (Figure 1). Details of the sediment sampling methods have been described previously [18,19] and are summarized here. Sediment samples were collected either by boat or while wading in the stream, targeting depositional areas with fine-grained sediments (silts). A push core sampler (WaterMark^®^ Universal Core Head Sediment Sampler; Forestry Suppliers Inc, Jackson, MS 39201, USA) with a polycarbonate tubing (Forestry Suppliers; 70 mm outer diameter × 66.7 mm inner diameter) was used to collect sediment from the surface up to a depth of 15 cm. The 15 cm depth was used to focus on recently deposited sediments. The sediment core was placed into a stainless-steel pan and divided vertically, and the halves were transferred to separate baked amber-glass jars. Samples were stored in the dark on ice, and within 48 h, they were shipped for chemical analyses. A new core tube was used at each sampling location. Between sampling locations, sediment processing equipment was cleaned using detergent (Alconox^®^, Alconox Inc., White Plains, NY 10603, USA) water followed by three rinses with tap water and three rinses with deionized water.

### 2.2. Chemical Analysis

The quantities of 208 different chemical pollutants, belonging to around sixteen broad chemical classes, including pesticides, industrial products, personal care products, and pharmaceuticals, were determined or estimated in water and sediment samples collected from eight different sites (Appendix A). Of the 208 chemicals measured in total, 175 were found in the water samples, 87 were found in the sediment samples, and 54 chemicals were common to both the water and sediment samples (Figure 2).

Details of the chemical analysis are described in prior publications [18,19] and are summarized here. Samples were transported to the laboratory for analysis under controlled conditions to prevent contamination or degradation. Sediment samples were dried, homogenized, and sieved, while water samples were filtered to remove particulate matter. Chemical contaminants were extracted from the samples using solvent extraction methods, followed by cleanup procedures to remove interfering substances. Gas Chromatography–Mass Spectrometry (GC-MS) (Agilent, Santa Clara, CA, 95051, USA) and Liquid Chromatography–Mass Spectrometry (LC-MS) (ThermoFisher Scientific, Waltham, MA, 02451, USA) were used for the detection and quantification of pollutants, providing high sensitivity and specificity for a wide range of chemical contaminants. Quality control and assurance measures included the analysis of calibration standards, blanks, and quality control samples alongside the environmental samples to ensure the reliability and reproducibility of the results. The concentration of pollutants was determined (in units of micrograms per kilograms of dry weight for the sediment samples and in nanograms per milliliter for the water samples) using calibration curves obtained from the standards, with data corrected for any losses during the extraction and cleanup processes.

### 2.3. Genomic Sample Processing

Genomic DNA from water samples was extracted using the Qiagen DNeasy^®^ PowerWater^®^ kit (QIAGEN, Germantown, MD, 20874, USA), following the manufacturer’s protocol. For the sediment samples, gDNA was extracted using the Qiagen DNeasy^®^ PowerSoil^®^ kit (QIAGEN, Germantown, MD, 20874, USA), following the manufacturer’s protocol. For both water and sediment samples, gDNA was eluted in 100 µL of DNase/RNase-free water. DNA was quantified using a NanoDrop 2000 (ThermoFisher Scientific, Waltham, MA, 02451, USA), and the presence of gDNA was verified using an Agilent DNA 1000 gel kit run on a Bioanalyzer 2100 (Agilent, Santa Clara, CA, 95051, USA). Purified gDNA was diluted in Resuspension Buffer (RSB) (Illumina, San Diego, CA, 92122, USA) to the required starting concentration for the TruSeq Nano DNA library preparation protocol (≈100 ng). The diluted DNA was then re-quantified using a NanoDrop 8000 Spectrophotometer (ThermoFisher Scientific, Waltham, MA, 02451, USA) and assessed for quality by Genomic DNA Screen Tapes using a TapeStation 4200 (Agilent, Santa Clara, CA, 95051, USA) to ensure dilution accuracy. Library preparations were performed according to the manufacturer’s instructions with minor insignificant modifications in MicroAmp 2 ml 8-Tube Strips (Applied Biosystems, Foster City, CA, 94404, USA). The quality and band size of libraries were assessed using D1000 Screen Tapes (Agilent, Santa Clara, CA, 95051, USA) on a Tapestation 4200 (Agilent). The amplified fragments were further quantified with a Qubit dsDNA HS Assay Kit (Invitrogen, Merelbeke, Belgium) on a Qubit 2.0 Fluorometer. Prior to sequencing, libraries were normalized to a working concentration range of 4.5–5.5 nM using the molarity calculated from Qubit. Libraries were pooled in groups of six per lane. The clustering of the index-coded samples was performed on a cBot Cluster Generation System using the HiSeq PE (Paired-End) Cluster Kit (Illumina, San Diego, CA, 92122, USA), according to the manufacturer’s instructions. Following cluster generation, the library preparations were sequenced on an Illumina HiSeq 4000 sequencer (Illumina, San Diego, CA, 92122, USA) using the HiSeq 3000/4000 SBS Kit (Illumina, San Diego, CA, 92122, USA). For sequencing, we used a single-indexed 151-read (paired-end sequencing) plan with a total of 325 cycles (151 bp reads, 8 bp index sequence, and 7 supplemental chemistry cycles). The resulting sequences were assessed and filtered according to base quality using the FASTQ Quality Filter (Hannon Lab, Cold Spring Harbor, NY, 11724, USA).

### 2.4. Data Analyses

Fastq files were trimmed using bbduk from the bbtools suite [20]. Trimmed reads were classified by taxonomy with the centrifuge tool from Johns Hopkins University [21], using the NCBI non-redundant nucleotide database. The downstream analysis of the taxonomy—such as alpha and beta diversity analyses—was performed with the phyloseq package in R v4.3.3 (https://www.r-project.org) (accessed on 20 January 2025). Correlation statistical analysis, including correlation and significance evaluations, were performed using the stats v4.3.2 [22], Hmisc v5.1-3 [23], psych v2.4.6.26 [24], corrplot v0.92 [25], car v3.1-2 [26], lsr v0.5.2 [27], multcomp v1.4-26 [28], and ggplot2 v3.5.1 [29] (ggpubr v0.0.0 [30] and GGally v2.2.1 [30]) packages in the R programming language.

### 2.5. Functional Pathway Annotation

Functional annotations of metagenomic reads were performed using HUMAnN3 (version 3.6) [31], which quantifies pathways and gene families from metagenomic data. Metabolic and xenobiotic degradation pathways were identified and quantified using the KEGG Orthology (KO) [32] database. Pathway-specific enzymatic activities, such as polycyclic aromatic hydrocarbon (PAH) oxygenases, bisphenol dioxygenases, and atrazine chlorohydrolases, were annotated based on their presence in the KO database and their relative abundance within each sample.

Pathways of interest, including “xenobiotic biodegradation and metabolism” (map00624), “alkaloid biosynthesis” (map00950), and “lipid metabolism” (map04979), were further analyzed to identify kingdom-specific contributions. Taxonomic contributors to each pathway were inferred by linking functional gene abundances to their respective microbial taxa using MetaPhlAn3 (version 3.1) [31].

### 2.6. Resistance Gene Identification

Resistance genes were identified using DeepARG (version 1.0) [33], which aligns metagenomic sequences against a curated antibiotic resistance gene (ARG) database. To ensure reliability, only ARGs with sequence identity ≥90% and alignment length ≥80% of the gene were retained. Gene abundances were normalized against the total number of reads per sample to account for sequencing depth variability.

## 3. Results

### 3.1. Chemical Concentrations per Site

Quantities of 175 and 87 chemical pollutants, spanning sixteen broad classes, were measured in the water and sediment samples, respectively, collected from eight distinct sites (Figure 2). The sediment samples had a much higher total concentration of chemical pollutants compared to the water samples (Figure 3). Sterols, PAHs, and detergent metabolites were the major classes of chemical pollutants found across the 8 sediment sampling sites (Figure 3a). Antioxidants were the most abundant class of pollutants at the UCJ sediment sampling site, which was unique compared to the other sites (Figure 3a). Overall, sterols were the most abundant class of chemicals in the water samples across all sampling sites (Figure 3b). Compared across sampling sites, fire retardants were most abundant in the samples collected from the MIE site (Figure 3b). Detergent metabolites, plasticizers, pharmaceuticals, insecticides, and flavors and fragrances were also dominant classes of chemicals in the water samples found across sites (Figure 3b).

### 3.2. Microbial Diversity and Relative Abundance

The total read counts for the sediment and water samples across the eight collection sites showed that the total sequence reads for microbiomes from the sediment samples have a larger dynamic range compared to those from the water samples (Appendix A and Table 1).

The alpha-diversity and Bray–Curtis distance per site for each sample type (sediment or water) were determined. Overall, the alpha-diversity in the sediment samples was higher (Figure 4a), whereas there was more dispersion along PC2 in the Bray–Curtis distance plot for the water samples (Figure 4b).

From the shotgun sequence reads, six kingdoms of microbiomes—bacteria, metazoa, viruses, fungi, protozoa, and archaea (listed according to their relative abundance)—were identified (Figure 5a). Nine bacterial phyla had a relative abundance >2%, with proteobacteria being the most abundant bacterial phylum, followed by actinobacteria (Figure 5b). Euryarchaeota had a relative abundance >2% in only two sediment samples (from the KKL and MET sites), and Nematoda had a relative abundance >2% in only one water sample (from the MEC site) (Figure 5b). Chordata, followed by Arthropoda, was the most abundant metazoan phyla (Figure 5c). Mollusca had a relative abundance >2% in only one water sample (from the MIM site) (Figure 5c).

Streptophyta, followed by chlorophyta, was the most abundant phyla of the viral kingdom (Figure 6a). Apicomplexa was the most abundant phylum of protozoan, except in the water sample from the KKL site, in which Ciliophora was the most abundant phylum (Figure 6b). Also, in the water sample from KKL, the phyla Haptista and Rhodophyta had relative abundances <2% (Figure 6b). The Discosea phylum had a relative abundance >2% in all of the sediment samples but was not found at >2% in any of the water samples (Figure 6b). Ascomycota and Basidomycota were the only two phyla of the fungal kingdom with a relative abundance >2% (Figure 6c).

### 3.3. Correlations Among Microbial Diversity and Levels (Concentrations) of Chemical Pollutants

Overall, the total number of sequence reads (assuming that it reflects the total microbiota density) showed a slight but significant negative correlation with concentration of chemical pollutants across all the sediment and water samples (Figure 7a). The analysis of the correlation of the different kingdoms vs. chemical pollutants across all the sampling sites and matrices revealed differing profiles depending on the kingdom. The abundance of the total viral population negatively correlated with the total concentration of chemical pollutants (R = −0.26, *p* < 2.2 × 10^−16^) (Figure 7b).

Similarly, the protozoan (R = −0.16, *p* < 2.2 × 10^−16^ and metazoan (R = −0.26, *p* < 2.2 × 10^−16^) populations were negatively correlated with the total concentrations of chemical pollutants (Figure 8a,c). Fungal population showed only a slightly negative but still significant correlation with the total concentration of chemical pollutants (Figure 8b).

On the other hand, bacterial or archaeal microbial populations were positively correlated with concentrations of chemical pollutants (R = 0.24, *p* < 2.2 × 10^−16^) (Figure 9a,b).

The abundance of the Euryarchaeota phylum was positively correlated with the total concentration of chemical pollutants (R = 0.14, 1.2 × 10^−7^) and with the concentrations of PAHs (R = 0.21, 3.9 × 10^−8^) in the sediment samples (Figure 10).

### 3.4. Microbial Community Correlations with Chemical Pollutants

The relationship between microbial community structures and chemical pollutant concentrations was investigated. A significant negative correlation was observed between the total microbiota density, estimated by the total number of sequence reads, and pollutant concentrations (Figure 7a). Viral and metazoan populations displayed the strongest negative correlation with pollutant concentrations (R = −0.26, *p* < 2.2 × 10^−16^) (Figure 7b and Figure 8c), followed by protozoa (R = −0.16, *p* < 2.2 × 10^−16^ (Figure 8a,c). Fungal population showed a weaker but still significant negative correlation (Figure 8b).

Contrastingly, bacterial and archaeal populations exhibited a significant positive correlation with pollutant concentrations (R = 0.24, *p* < 2.2 × 10^−16^) (Figure 9a,b). Notably, the archaeal Euryarchaeota phylum showed positive correlations with the total pollutant concentrations (R = 0.14, *p* < 1.2 × 10^−7^) and PAHs (R = 0.21, *p* < 3.9 × 10^−8^) in the sediment samples (Figure 10a,b).

Overall, the concentrations of chemical pollutants from sediment samples were much higher and had a larger dynamic range than in the water samples, though the water samples had a greater diversity of chemical species. The total microbiota density (as approximated from the adjusted and normalized total counts of sequence reads) decreased with an increased total concentration of chemical pollutants. Protozoan, metazoan, and fungal populations were negatively correlated with the concentrations of chemical pollutants, whereas some bacterial (Proteobacteria) and archaeal (Euryarchaeota) populations were positively correlated with increasing concentrations of chemical pollutants (such as PAHs).

### 3.5. Enzymatic and Pathway Associations with Microbial Kingdoms

This study identified enzymes and pathways linked to pollutant biodegradation across microbial kingdoms. For bacteria and archaea, pathways such as xenobiotics biodegradation (map00624), lipid metabolism (map04979), and the metabolism of secondary metabolites (map00403) were prominent (Table 2). Enzymes such as catechol 2,3-dioxygenase, bisphenol dioxygenase, and cholesterol oxidase were associated with these pathways, supporting their roles in chemical pollutant breakdown (Table 2).

Fungi were associated with pathways like indole diterpene alkaloid biosynthesis (map00403) and xenobiotics biodegradation (map00624), using enzymes such as laccase and peroxidase (Table 2). These pathways suggest their involvement in the degradation of complex aromatic compounds, although their efficiency diminishes at higher pollutant concentrations.

### 3.6. Resistance Gene Detection

The detection of resistance genes in the analyzed dataset highlights microbial adaptations to environmental chemical pollutants and their potential roles in biodegradation processes. Functional annotations, pathway identifications, and enzyme activity profiling led to the results shown in Table 3.

## 4. Discussion

This study investigated the diversity and abundance of microbial communities in pollutant-impacted sediment and freshwater ecosystems and correlated them with the corresponding levels of anthropogenic chemical pollutants. The combination of anthropogenic and natural effects led to unique patterns of microbial diversity across the different locations. The findings of this study provided important insights into the differential distribution and effect of chemical pollutants in the sediment and water samples and their subsequent effects on microbial communities in polluted environments.

The results reveal that sediment samples contain significantly higher concentrations of chemicals compared to water samples (Figure 3). This result is consistent with the existing literature, which suggests that pollutants tend to be adsorbed into sediment particles, leading to higher pollutant concentrations in sediments than in water [18]. The major classes of chemical pollutants identified in the sediments include sterols, polycyclic aromatic hydrocarbons (PAHs), and detergent metabolites, while sterols were also predominant in the water samples across all sites (Figure 3a,b). These observations underscore the persistence and bio-accumulative properties of these compounds [37].

Different pollutant profiles were observed at specific sites, such as the predominance of antioxidants at the UCJ sediment site and the high levels of fire retardants in the water samples from the MIE site (Figure 3a,b). These site-specific variations highlight the importance of identifying local pollution sources to inform targeted mitigation strategies. The detection of high concentrations of fire retardants at the MIE site, for example, points to localized sources of contamination, possibly from industrial activities or waste discharges associated with fire retardant use [38].

The microbial diversity was higher in the sediment samples than in the water samples, as evidenced by the greater alpha-diversity and broader dynamic range of total sequence reads (Figure 4a and Appendix A and Table 1). This higher diversity in sediments can be attributed to the complex and heterogeneous nature of sediments, which provide diverse microhabitats for microorganisms [39]. In contrast, the greater dispersion along PC2 in the Bray–Curtis distance plot for the water samples suggests more variability in microbial community composition across the different water sites (Figure 4b) [40].

The microbial community analysis identified six kingdoms, with bacteria being the most abundant, followed by metazoa, viruses, fungi, protozoa, and archaea (Figure 5a). Previous research has shown that Proteobacteria, the most common bacterial phylum, are involved in natural biogeochemical processes and can perform anaerobic degradation of organic pollutants in groundwater basins [41]. The positive correlation between bacterial and archaeal populations and the concentration of chemical pollutants suggests that these microbial groups may possess mechanisms to tolerate or metabolize certain pollutants (Figure 9a,b). This finding aligns with studies showing that some bacterial and archaeal taxa can thrive in contaminated environments by using pollutants as energy sources [42].

Conversely, the negative correlation between the abundance of viral, protozoan, and fungal populations and the concentration of chemical pollutants (Figure 7b and Figure 8a,c) indicates that these groups are more susceptible to pollution stress. This decline in susceptible populations could disrupt ecological interactions and the functioning of microbial communities, ultimately affecting ecosystem health [43]. The correlation between the abundance of Euryarchaeota and the concentration of chemical pollutants, particularly PAHs, underscores the role of specific microbial taxa in pollutant degradation and adaptation to contaminated environments (Figure 10) [44]. These findings suggest that certain microbial taxa can serve as biomarkers for pollution monitoring and bioremediation efforts [45].

### 4.1. Role of Microbial Kingdoms in Pollutant Degradation

#### 4.1.1. Bacteria and Archaea

The interactions between microbial communities and environmental chemical pollutants provide insights into the effects of chemical pollutants on diverse microbial kingdoms. The positive correlations between bacterial and archaeal populations and pollutant concentrations reflect their adaptability and metabolic versatility. Key pathways, such as the biodegradation of xenobiotics (map00624), enable these kingdoms to metabolize complex pollutants like polycyclic aromatic hydrocarbons (PAHs). Proteobacteria, often dominant in polluted environments, harbor enzymes such as catechol 2,3-dioxygenase and bisphenol dioxygenase, which are important for breaking down aromatic hydrocarbons [46,47]. Archaeal species, including those from the Euryarchaeota phylum, contribute significantly to PAH degradation under anaerobic conditions by using enzymes such as PAH dioxygenases and naphthalene dioxygenase [48,49]. Additionally, bacterial genes encoding atrazine chlorohydrolase are important in herbicide degradation and agricultural runoff remediation [50].

#### 4.1.2. Fungi

Fungi demonstrate their ecological significance in pollutant detoxification through enzymatic systems such as laccases and peroxidases. These enzymes degrade aromatic compounds and enhance oxidative stress resistance, indicating fungi’s dual roles in survival and pollutant biodegradation. Fungal peroxidases, for instance, have been implicated in the breakdown of a range of environmental chemical pollutants [51]. However, the reduced abundance of fungi at higher pollutant concentrations suggests potential inhibitory effects on fungal metabolism, necessitating strategies to enhance fungal resilience [52]. Combining fungi with bacterial and archaeal consortia could significantly improve pollutant degradation efficiency.

#### 4.1.3. Protozoa, Viruses, and Metazoans

Protozoa, reliant on bacterial prey, often experience indirect effects from shifts in bacterial community composition induced by environmental pollutants. Stressors such as chemical pollutants can alter protozoan community structures and functional roles [53,54]. Additionally, protozoa demonstrate potential pollutant biotransformation capabilities through dioxygenases for aromatic hydrocarbon degradation, highlighting their emerging ecological roles [55].

Viruses indirectly influence pollutant degradation through host-derived enzymes and viral-mediated horizontal gene transfer, enhancing microbial degradation potential. Environmental pollutants, however, may inhibit viral replication and influence microbial community dynamics. Studies suggest that viruses play an integral role in transferring metabolic genes, thereby impacting microbial community structure and function [56,57].

Metazoans, including other soil invertebrates, contribute indirectly to pollutant distribution and microbial activity through sediment bioturbation. These activities influence the degradation of pollutants like PAHs, with bioturbation by benthic organisms being shown to enhance microbial degradation in sediments [49,58]. However, pollution can adversely affect metazoan health and biodiversity, leading to disruptions in ecosystem dynamics and functions [59].

### 4.2. Implications for Ecosystem Restoration

Bacteria and archaea, as primary degraders, can be leveraged alongside fungi to enhance pollutant breakdown. Protecting sensitive groups such as protozoa and viruses is critical to maintaining ecological balance and optimizing bioremediation efficiency. Bioaugmentation with pollutant-degrading strains, along with advancements in genetic engineering, could further optimize microbial pathways, enabling the tailored degradation of specific pollutants [60]. For instance, (i) the degradation of herbicides and pesticides by bacterial and fungal enzymes could mitigate agrochemical impacts on soil and water quality. (ii) Archaeal enzymes involved in PAH degradation could serve as biomarkers for monitoring bioremediation efficacy in contaminated sediments.

### 4.3. Limitations and Future Directions

While this study highlights the significant roles of resistance genes and enzymatic pathways, future research should (i) determine if other stressors (such as variations in temperature, pH, nutrient level) have impacts and assess how the different types of stressors might interact; (ii) develop predictive models that integrate microbial community dynamics and pollutant degradation pathways for optimized bioremediation strategies; and (iii) investigate virus–host interactions to elucidate their contributions to resistance gene dissemination and pollutant degradation.

Finally, this study provides insights into the distribution and impact of chemical pollutants on microbial communities in sediment and water samples. The observed correlations between pollutant concentrations and microbial diversity emphasize the need for ongoing monitoring and targeted strategies to mitigate the effects of pollution on ecosystems. The findings of this study support the utility of microbiome diversity in environmental samples to distinguish between the concentration gradients of chemical pollutants [61]. The comprehensive characterization of microbial communities and their responses to pollutants can aid in developing biomarkers for environmental health and enhancing bioremediation strategies [12]. Future research should continue to explore the mechanisms underlying microbial responses to pollutants to inform more effective environmental management and conservation practices [13].

## 5. Conclusions

This study demonstrates that sediment harbors higher levels of anthropogenic pollutants compared to overlying water, contributing to distinct and more diverse microbial communities. Bacteria and archaea exhibited positive correlations with elevated pollutant concentrations, reflecting their capacity for metabolic adaptation and their potential to degrade complex compounds such as polycyclic aromatic hydrocarbons. Conversely, the reduced abundances of viruses, protozoa, and fungi at higher pollution levels highlight the vulnerability of these groups and underscore the importance of protecting sensitive taxa to maintain ecological stability. By identifying key microbial populations and specific enzymatic pathways linked to pollutant degradation, the findings support the use of microbiome diversity as a biomarker for environmental monitoring. Furthermore, the results indicate that strategic bioremediation approaches—such as leveraging bacterial, fungal, and archaeal consortia—could enhance pollutant breakdown and facilitate ecological restoration. Future work should focus on developing predictive models to guide remediation strategies and elucidating virus–host interactions to fully harness the potential of microbial communities in mitigating environmental pollution.

## Figures and Tables

**Figure 1 toxics-13-00142-f001:**
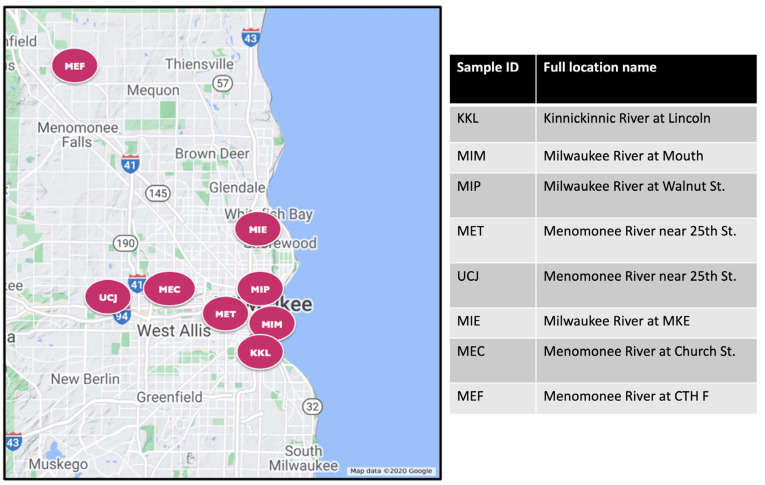
Map showing sites of collection for sediment and water samples. The expanded name of each site is shown on the right panel.

**Figure 2 toxics-13-00142-f002:**
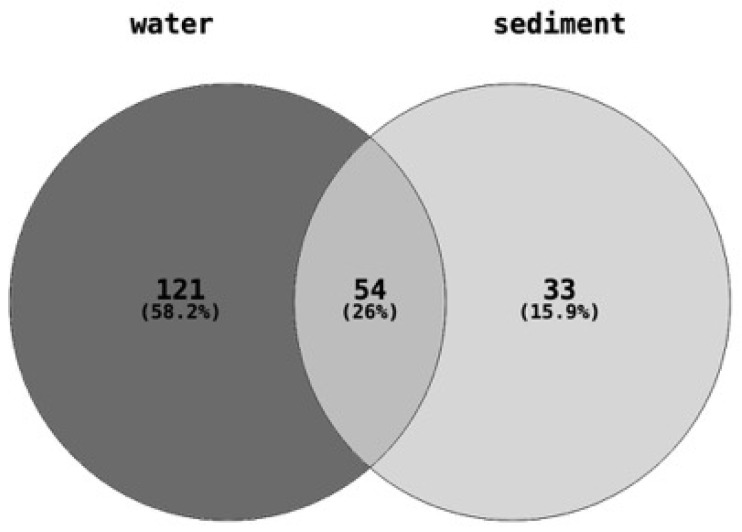
Number of chemical pollutants quantified in each of the water (175 chemicals) and sediment (87 chemicals) samples. A total of 208 chemical contaminants belonging to pharmaceuticals and personal care products, plasticizers or flame retardants, fuels and polyaromatic hydrocarbons, wastewater indicators (sterols), and other industrial chemicals were estimated or measured in the water and sediment samples collected from each of the eight sites.

**Figure 3 toxics-13-00142-f003:**
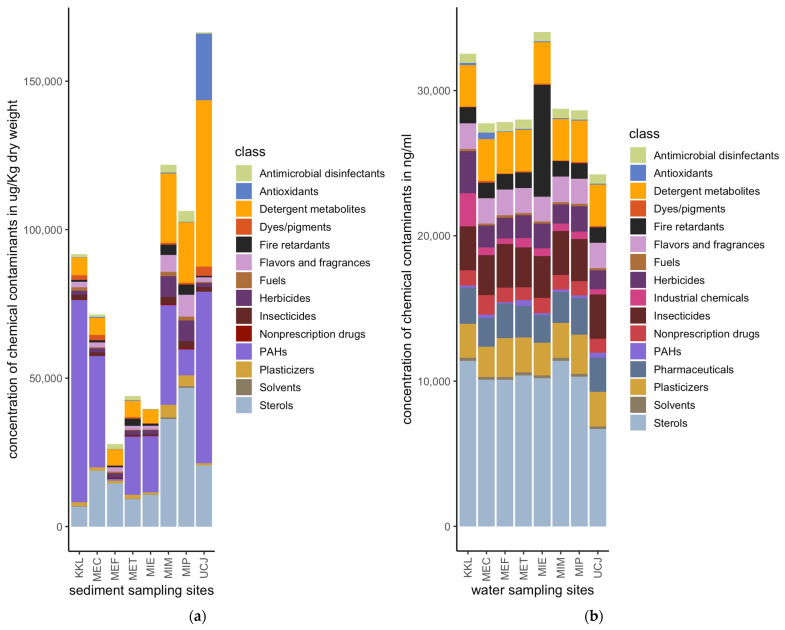
Relative amounts of chemical pollutants per site for water or sediment samples: (**a**) Concentrations of classes of chemical pollutants at the sediment sampling sites. (**b**) Concentrations of classes of chemical pollutants at the water sampling sites. Sediment samples have a much higher total concentration of chemical pollutants compared to water samples. Key: PAHs: polycyclic aromatic hydrocarbons.

**Figure 4 toxics-13-00142-f004:**
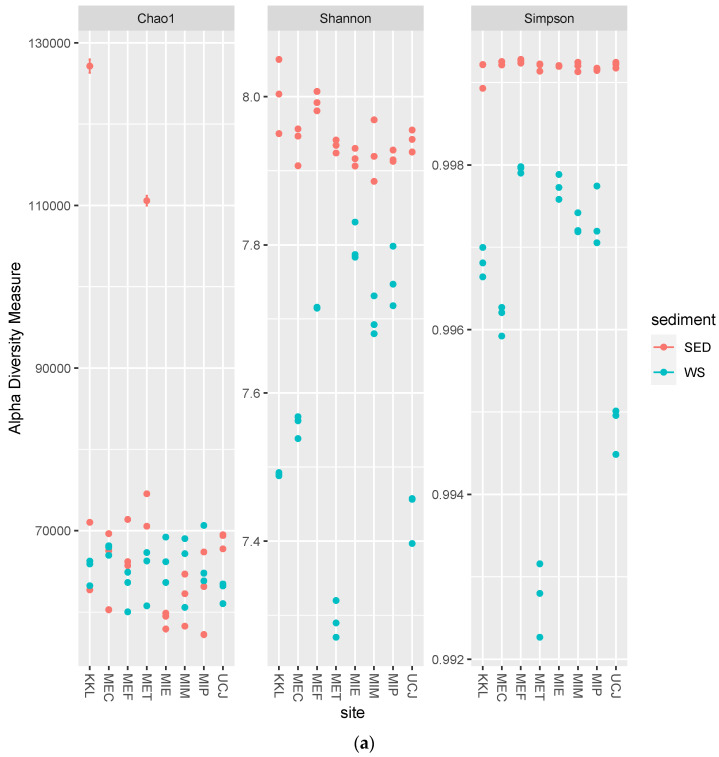
Alpha-diversity and Bray–Curtis distance per site for each sample type (sediment or water) (**a**) Alpha-diversity per site. The alpha-diversity in the sediment samples is higher; (**b**) Bray–Curtis distance per site. Water samples showed more dispersion along PC2 and were clustered by site. Keys: SED—sediment; WS—water.

**Figure 5 toxics-13-00142-f005:**
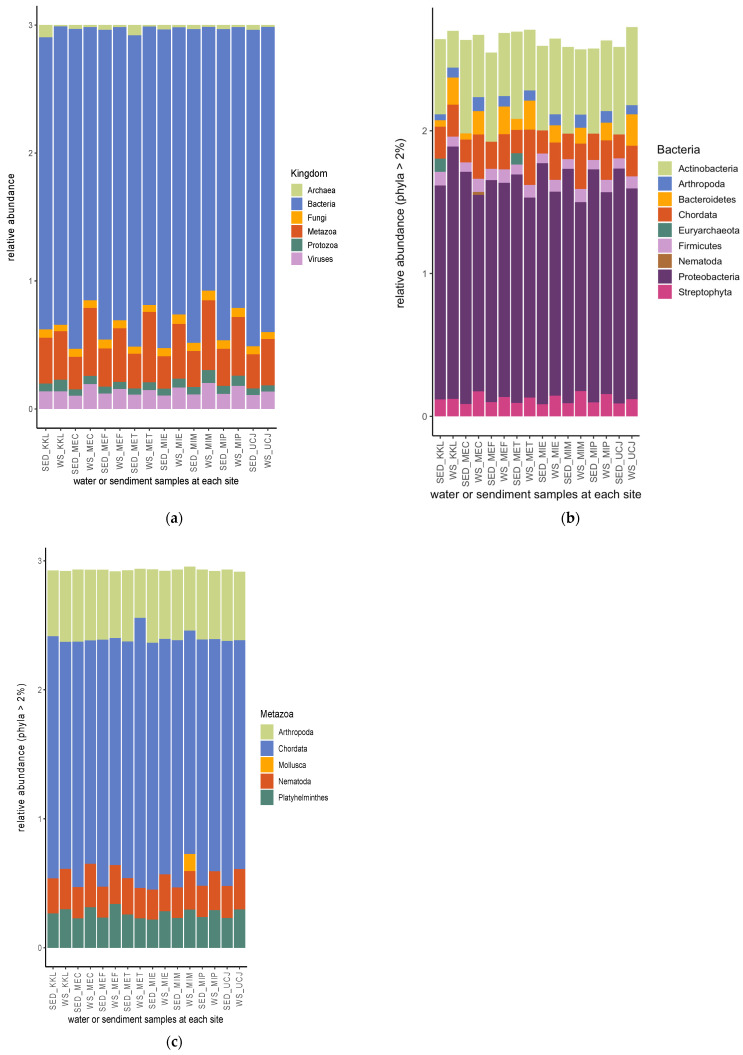
Relative abundances of microbiomes’ (**a**) kingdom and (**b**) bacterial and (**c**) metazoan phyla. Bacteria are the most abundant among the microbiota kingdom, and proteobacteria and chordata are the most abundant bacterial and metazoan phyla, respectively. Keys: SED—sediment; WS—water.

**Figure 6 toxics-13-00142-f006:**
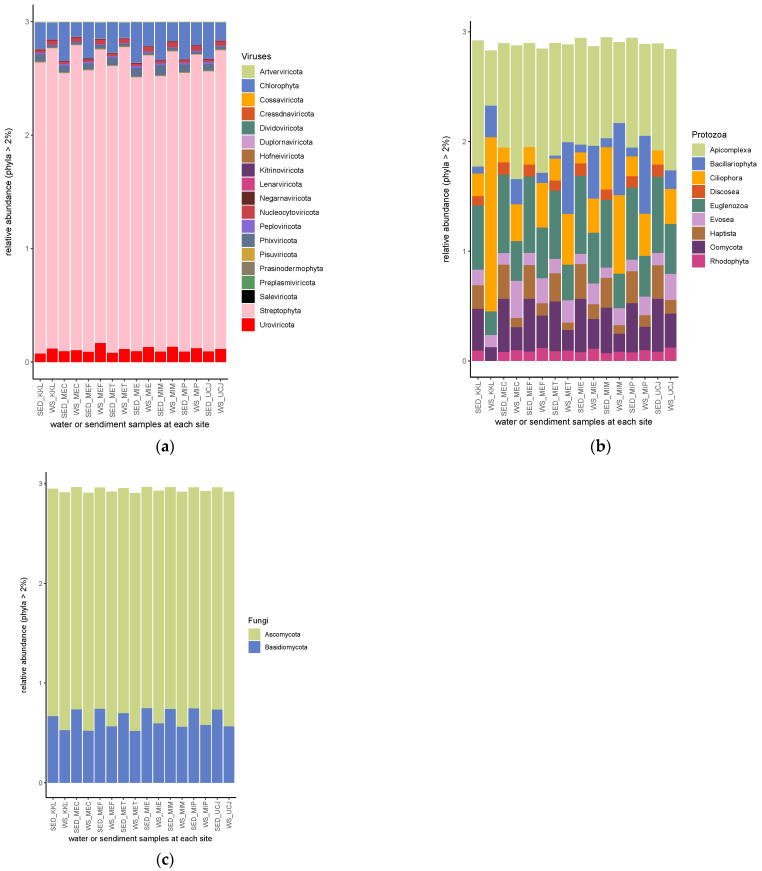
Relative abundances of (**a**) viral, (**b**) protozoan, and (**c**) fungal phyla. Streptophyta, apicomplexa, and ascomycota are the most abundant among the phyla of viral, protozoan, and fungal phyla, respectively. SED—sediment; WS—water.

**Figure 7 toxics-13-00142-f007:**
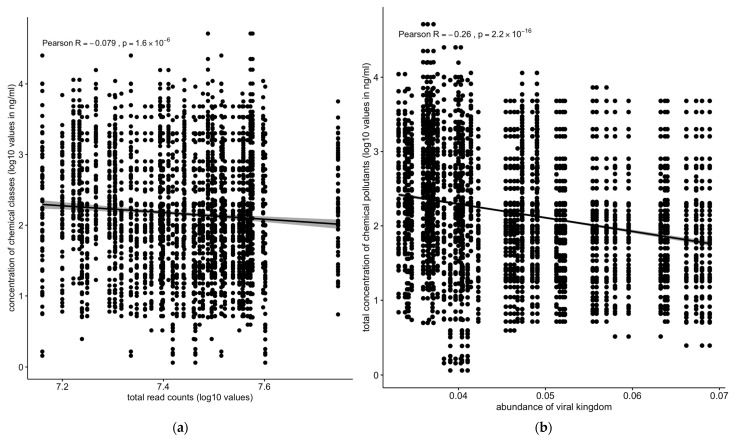
Correlation between concentrations of chemical pollutants vs. (**a**) total read counts or (**b**) viral kingdom. The total number of sequences reads (if we assume that this reflects the total microbiota density) was negatively correlated with the concentration of chemical pollutants: the abundance of the total viral population negatively correlated with the total concentration of chemical pollutants. Solid line: linear regression fit; shaded area around the solid line: 95% confidence interval; the dots: are scatter points showing the relationship between total read counts (**a**) or abundance of viral kingdom (**b**) and chemical concentration.

**Figure 8 toxics-13-00142-f008:**
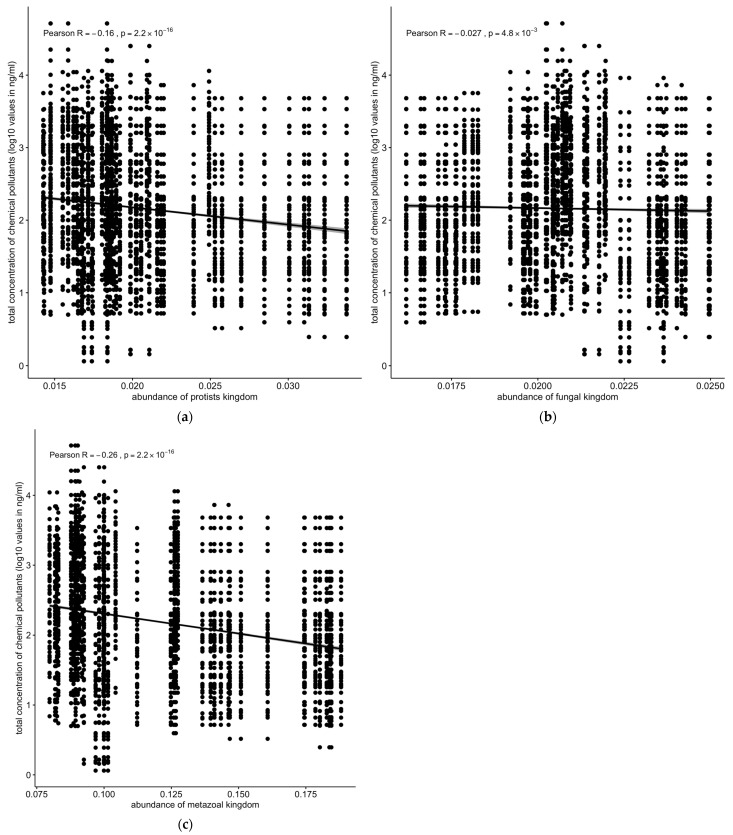
Correlation between concentrations of chemical pollutants vs. (**a**) protozoan, (**b**) fungal, or (**c**) metazoan microbial populations. Protozoan, metazoan, and fungal populations were negatively correlated with total concentrations of chemical pollutants. Solid line: linear regression fit; shaded area around the solid line: 95% confidence interval; the dots: are scatter points showing the relationship between abundance of microbial kingdom and chemical concentration.

**Figure 9 toxics-13-00142-f009:**
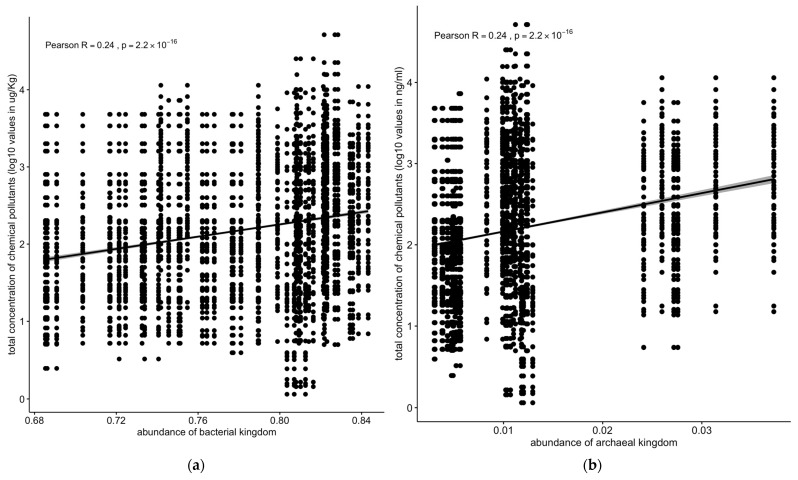
Correlation between concentrations of chemical pollutants vs. (**a**) bacterial or (**b**) archaeal microbial populations. The correlation stats for bacterial and archaeal kingdoms are identical. The data were verified (starting from the raw data), and these are the correct values. Solid line: linear regression fit; shaded area around the solid line: 95% confidence interval; the dots: are scatter points showing the relationship between abundance of microbial kingdom and chemical concentration.

**Figure 10 toxics-13-00142-f010:**
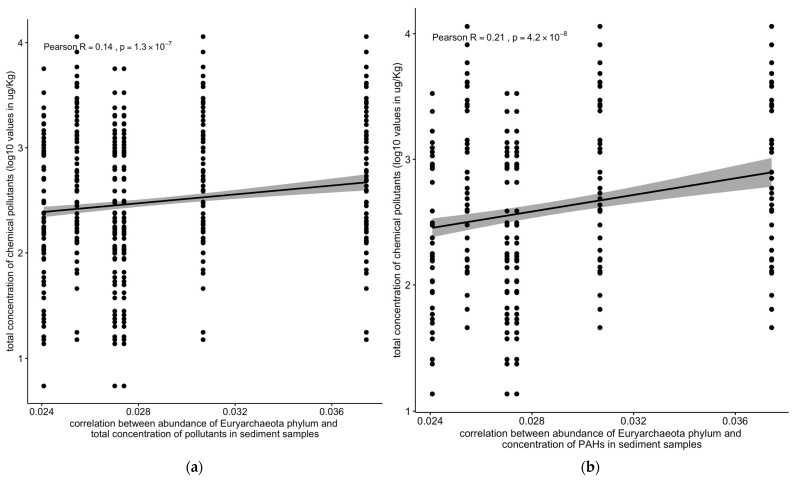
Correlation between the abundance of the Euryarchaeota phylum vs. (**a**) the total concentration of chemical pollutants or (**b**) polycyclic aromatic hydrocarbons (PAHs). Solid line: linear regression fit; shaded area around the solid line: 95% confidence interval; the dots: are scatter points showing the relationship between the abundance of the Euryarchaeota phylum and total chemical concentration or PAHs.

**Table 1 toxics-13-00142-t001:** Total sequence reads for microbiomes from sediment and water samples. Total reads from sediments have larger dynamic range compared to those from water samples.

Site	Sediment Read Counts	Water Read Counts
KKL	20,476,097.33	34,616,707.33
MEC	30,385,105.33	33,736,439.67
MEF	31,737,857.00	21,904,403.33
MET	41,576,421.67	32,134,992.33
MIE	18,537,225.33	28,940,417.67
MIM	22,971,062.33	25,820,810.67
MIP	22,941,852.00	28,089,634.00
UCJ	35,277,928.33	25,489,005.00

**Table 2 toxics-13-00142-t002:** Microbial taxa and enzymatic pathways involved in the biodegradation of diverse environmental chemical pollutants.

OTUTAXAID	Kingdom	Class of Chemical Pollutants	Chemical	Class	Pid	Enzyme	Pathway_Name	Enzyme_ID	Orthology
OTU57611	Archaea	Antioxidants	Bisphenol degradation	Metabolism of xenobiotics	map00363	Antioxidant reductase	Bisphenol degradation	Catechol 2,3-dioxygenase (EC 1.13.11.2)	K00537 [34]
OTU57611	Archaea	Antimicrobial disinfectants	Bisphenol degradation	Metabolism of xenobiotics	map00363	Bisphenol dioxygenase	Bisphenol degradation	Nicotine dehydrogenase	K00536 [35]
OTU57611	Archaea	Flavors and fragrances	Indole diterpene alkaloid biosynthesis	Biosynthesis of secondary metabolites	map00403	Indole biosynthase	Indole diterpene alkaloid biosynthesis	Laccase	K05033 [36]
OTU57611	Archaea	PAHs	Polycyclic aromatic hydrocarbon degradation	Xenobiotics biodegradation	map00624	PAH oxygenase	Polycyclic aromatic hydrocarbon degradation	Naphthalene dioxygenase (EC 1.14.12.12)	K14579 [35]
OTU57611	Archaea	Herbicides	Atrazine degradation	Biodegradation of herbicides	map00791	Atrazine hydrolase	Atrazine degradation	Atrazine chlorohydrolase	K14580 [34]
OTU57611	Archaea	Insecticides	Atrazine degradation	Biodegradation of herbicides	map00791	Atrazine hydrolase	Atrazine degradation	Isoquinoline dioxygenase	K14581 [36]
OTU57611	Archaea	Solvents	Isoquinoline alkaloid biosynthesis	Alkaloid biosynthesis	map00950	Quinoline oxidase	Isoquinoline alkaloid biosynthesis	Cholesterol oxidase (EC 1.1.3.6)	K00536 [35]
OTU57611	Archaea	Sterols	Cholesterol metabolism	Lipid metabolism	map04979	Cholesterol demethylase	Cholesterol metabolism	Nicotine dehydrogenase	K00537 [34]
OTU57611	Archaea	Nonprescription drugs	Nicotine addiction	Drug addiction pathways	map05033	Monoamine oxidase	Nicotine addiction	Catechol 2,3-dioxygenase (EC 1.13.11.2)	K05033 [36]
OTU65079	Bacteria	Antimicrobial disinfectants	Bisphenol degradation	Metabolism of xenobiotics	map00363	Bisphenol oxidase	Bisphenol degradation	Catechol 2,3-dioxygenase (EC 1.13.11.2)	K14579 [35]
OTU65079	Bacteria	Antioxidants	Bisphenol degradation	Metabolism of xenobiotics	map00363	Peroxidase	Bisphenol degradation	Laccase	K14580 [34]
OTU65079	Bacteria	Flavors and fragrances	Indole diterpene alkaloid biosynthesis	Biosynthesis of secondary metabolites	map00403	Indole alkaloid synthase	Indole diterpene alkaloid biosynthesis	Naphthalene dioxygenase (EC 1.14.12.12)	K14581 [36]
OTU65079	Bacteria	PAHs	Polycyclic aromatic hydrocarbon degradation	Xenobiotics biodegradation	map00624	PAH dioxygenase	Polycyclic aromatic hydrocarbon degradation	Atrazine chlorohydrolase	K00536 [35]
OTU65079	Bacteria	Insecticides	Atrazine degradation	Biodegradation of herbicides	map00791	Atrazine dechlorinase	Atrazine degradation	Cholesterol oxidase (EC 1.1.3.6)	K05033 [34]
OTU65079	Bacteria	Herbicides	Atrazine degradation	Biodegradation of herbicides	map00791	Atrazine chlorohydrolase	Atrazine degradation	Isoquinoline dioxygenase	K00537 [36]
OTU65079	Bacteria	Solvents	Isoquinoline alkaloid biosynthesis	Alkaloid biosynthesis	map00950	Monooxygenase	Isoquinoline alkaloid biosynthesis	Nicotine dehydrogenase	K14579 [35]
OTU65079	Bacteria	Sterols	Cholesterol metabolism	Lipid metabolism	map04979	Cholesterol oxidase	Cholesterol metabolism	Catechol 2,3-dioxygenase (EC 1.13.11.2)	K14580 [34]
OTU65079	Bacteria	Nonprescription drugs	Nicotine addiction	Drug addiction pathways	map05033	Monoamine oxidase	Nicotine addiction	Laccase	K14581 [36]
OTU39239	Fungi	Antioxidants	Bisphenol degradation	Metabolism of xenobiotics	map00363	Peroxidase	Bisphenol degradation	Isoquinoline dioxygenase	K00537 [35]
OTU39239	Fungi	Antimicrobial disinfectants	Bisphenol degradation	Metabolism of xenobiotics	map00363	Laccase	Bisphenol degradation	Atrazine chlorohydrolase	K00536 [34]
OTU39239	Fungi	Flavors and fragrances	Indole diterpene alkaloid biosynthesis	Biosynthesis of secondary metabolites	map00403	Dioxygenase	Indole diterpene alkaloid biosynthesis	Cholesterol oxidase (EC 1.1.3.6)	K05033 [36]
OTU39239	Fungi	PAHs	Polycyclic aromatic hydrocarbon degradation	Xenobiotics biodegradation	map00624	PAH hydroxylase	Polycyclic aromatic hydrocarbon degradation	Nicotine dehydrogenase	K14579 [35]
OTU39239	Fungi	Insecticides	Atrazine degradation	Biodegradation of herbicides	map00791	Atrazine dechlorinase	Atrazine degradation	Laccase	K14581 [34]
OTU39239	Fungi	Herbicides	Atrazine degradation	Biodegradation of herbicides	map00791	Atrazine chlorohydrolase	Atrazine degradation	Catechol 2,3-dioxygenase (EC 1.13.11.2)	K14580 [36]
OTU39239	Fungi	Solvents	Isoquinoline alkaloid biosynthesis	Alkaloid biosynthesis	map00950	Monooxygenase	Isoquinoline alkaloid biosynthesis	Naphthalene dioxygenase (EC 1.14.12.12)	K00536 [35]
OTU39239	Fungi	Sterols	Cholesterol metabolism	Lipid metabolism	map04979	Cholesterol oxidase	Cholesterol metabolism	Atrazine chlorohydrolase	K00537 [34]
OTU39239	Fungi	Nonprescription drugs	Nicotine addiction	Drug addiction pathways	map05033	Monoamine oxidase	Nicotine addiction	Isoquinoline dioxygenase	K05033 [36]
OTU4082	Metazoa	Antimicrobial disinfectants	Bisphenol degradation	Metabolism of xenobiotics	map00363	Cytochrome P450	Bisphenol degradation	Naphthalene dioxygenase (EC 1.14.12.12)	K00536 [35]
OTU4082	Metazoa	Antioxidants	Bisphenol degradation	Metabolism of xenobiotics	map00363	Glutathione S-transferase	Bisphenol degradation	Atrazine chlorohydrolase	K00537 [34]
OTU4082	Metazoa	Flavors and fragrances	Indole diterpene alkaloid biosynthesis	Biosynthesis of secondary metabolites	map00403	Alkaloid demethylase	Indole diterpene alkaloid biosynthesis	Isoquinoline dioxygenase	K05033 [36]
OTU4082	Metazoa	PAHs	Polycyclic aromatic hydrocarbon degradation	Xenobiotics biodegradation	map00624	Aryl hydrocarbon hydroxylase	Polycyclic aromatic hydrocarbon degradation	Cholesterol oxidase (EC 1.1.3.6)	K14579 [35]
OTU4082	Metazoa	Herbicides	Atrazine degradation	Biodegradation of herbicides	map00791	Dehalogenase	Atrazine degradation	Nicotine dehydrogenase	K14580 [34]
OTU4082	Metazoa	Insecticides	Atrazine degradation	Biodegradation of herbicides	map00791	Dehalogenase	Atrazine degradation	Catechol 2,3-dioxygenase (EC 1.13.11.2)	K14581 [36]
OTU4082	Metazoa	Solvents	Isoquinoline alkaloid biosynthesis	Alkaloid biosynthesis	map00950	Aldehyde oxidase	Isoquinoline alkaloid biosynthesis	Laccase	K00536 [35]
OTU4082	Metazoa	Sterols	Cholesterol metabolism	Lipid metabolism	map04979	Cholesterol 7-alpha-hydroxylase	Cholesterol metabolism	Naphthalene dioxygenase (EC 1.14.12.12)	K00537 [34]
OTU4082	Metazoa	Nonprescription drugs	Nicotine addiction	Drug addiction pathways	map05033	MAO (Monoamine oxidase)	Nicotine addiction	Atrazine chlorohydrolase	K05033 [36]
OTU1565	Protozoa	Antioxidants	Bisphenol degradation	Metabolism of xenobiotics	map00363	Antioxidant reductase	Bisphenol degradation	Naphthalene dioxygenase (EC 1.14.12.12)	K14580 [35]
OTU1565	Protozoa	Antimicrobial disinfectants	Bisphenol degradation	Metabolism of xenobiotics	map00363	Oxidase	Bisphenol degradation	Laccase	K14579 [34]
OTU1565	Protozoa	Flavors and fragrances	Indole diterpene alkaloid biosynthesis	Biosynthesis of secondary metabolites	map00403	Indole synthase	Indole diterpene alkaloid biosynthesis	Atrazine chlorohydrolase	K14581 [36]
OTU1565	Protozoa	PAHs	Polycyclic aromatic hydrocarbon degradation	Xenobiotics biodegradation	map00624	Dioxygenase	Polycyclic aromatic hydrocarbon degradation	Isoquinoline dioxygenase	K00536 [35]
OTU1565	Protozoa	Insecticides	Atrazine degradation	Biodegradation of herbicides	map00791	Dehalogenase	Atrazine degradation	Nicotine dehydrogenase	K05033 [34]
OTU1565	Protozoa	Herbicides	Atrazine degradation	Biodegradation of herbicides	map00791	Hydrolase	Atrazine degradation	Cholesterol oxidase (EC 1.1.3.6)	K00537 [36]
OTU1565	Protozoa	Solvents	Isoquinoline alkaloid biosynthesis	Alkaloid biosynthesis	map00950	Monooxygenase	Isoquinoline alkaloid biosynthesis	Catechol 2,3-dioxygenase (EC 1.13.11.2)	K14579 [35]
OTU1565	Protozoa	Sterols	Cholesterol metabolism	Lipid metabolism	map04979	Oxidase	Cholesterol metabolism	Laccase	K14580 [34]
OTU1565	Protozoa	Nonprescription drugs	Nicotine addiction	Drug addiction pathways	map05033	Amine oxidase	Nicotine addiction	Naphthalene dioxygenase (EC 1.14.12.12)	K14581 [36]
OTU2161	Viruses	Antimicrobial disinfectants	Bisphenol degradation	Metabolism of xenobiotics	map00363	(Host-derived enzymes)	Bisphenol degradation	Isoquinoline dioxygenase	K14579 [35]
OTU2161	Viruses	Antioxidants	Bisphenol degradation	Metabolism of xenobiotics	map00363	(Host-derived enzymes)	Bisphenol degradation	Cholesterol oxidase (EC 1.1.3.6)	K14580 [34]
OTU2161	Viruses	Flavors and fragrances	Indole diterpene alkaloid biosynthesis	Biosynthesis of secondary metabolites	map00403	(Host-derived enzymes)	Indole diterpene alkaloid biosynthesis	Nicotine dehydrogenase	K14581 [36]
OTU2161	Viruses	PAHs	Polycyclic aromatic hydrocarbon degradation	Xenobiotics biodegradation	map00624	(Host-derived enzymes)	Polycyclic aromatic hydrocarbon degradation	Catechol 2,3-dioxygenase (EC 1.13.11.2)	K00536 [35]
OTU2161	Viruses	Herbicides	Atrazine degradation	Biodegradation of herbicides	map00791	(Host-derived enzymes)	Atrazine degradation	Laccase	K00537 [34]
OTU2161	Viruses	Insecticides	Atrazine degradation	Biodegradation of herbicides	map00791	(Host-derived enzymes)	Atrazine degradation	Naphthalene dioxygenase (EC 1.14.12.12)	K05033 [36]
OTU2161	Viruses	Solvents	Isoquinoline alkaloid biosynthesis	Alkaloid biosynthesis	map00950	(Host-derived enzymes)	Isoquinoline alkaloid biosynthesis	Atrazine chlorohydrolase	K14579 [35]
OTU2161	Viruses	Sterols	Cholesterol metabolism	Lipid metabolism	map04979	(Host-derived enzymes)	Cholesterol metabolism	Isoquinoline dioxygenase	K14580 [34]
OTU2161	Viruses	Nonprescription drugs	Nicotine addiction	Drug addiction pathways	map05033	(Host-derived enzymes)	Nicotine addiction	Cholesterol oxidase (EC 1.1.3.6)	K14581 [36]

**Table 3 toxics-13-00142-t003:** Resistance genes by microbial kingdom and chemical pollutant class.

Kingdom	Class of Chemical Pollutant	Enzyme Detected	Resistance Mechanism	Orthology ID	Pathway
Archaea	Antimicrobial Disinfectants	Bisphenol dioxygenase	Xenobiotic metabolism and detoxification	K00536	Bisphenol degradation [35]
Archaea	PAHs	PAH oxygenase	PAH degradation and detoxification	K14579	Polycyclic aromatic hydrocarbon degradation [35]
Bacteria	Herbicides	Atrazine chlorohydrolase	Herbicide degradation	K14580	Atrazine degradation [34]
Bacteria	Sterols	Cholesterol oxidase	Cholesterol metabolism and detoxification	K14580	Cholesterol metabolism [34]
Fungi	Antioxidants	Peroxidase	Oxidative stress resistance	K00537	Bisphenol degradation [34]
Fungi	Insecticides	Atrazine dechlorinase	Xenobiotic metabolism	K14581	Atrazine degradation [36]
Protozoa	PAHs	Dioxygenase	Aromatic hydrocarbon degradation	K00536	Polycyclic aromatic hydrocarbon degradation [35]
Viruses	Solvents	Host-derived enzymes	Isoquinoline alkaloid biosynthesis via hosts	K14579	Isoquinoline alkaloid biosynthesis [35]

## Data Availability

Datasets will be available on the journal’s website as Appendix A.

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
