# Peer review of "Effects of Environmental Chemical Pollutants on Microbiome Diversity: Insights from Shotgun Metagenomics"

_toxics, 2025, doi:10.3390/toxics13020142_

Round 1

Reviewer 1 Report

Comments and Suggestions for Authors

The manuscript is very interesting, as the results provide important information on the differential distribution and effects of chemical pollutants in sediment and water samples, and their subsequent effects on microbial communities in polluted environments. Understanding the dynamic responses of microbial communities to chemical contamination is essential for predicting potential long-term changes in ecosystems. Consequently, such an approach can help develop biomarkers for environmental health and improve bioremediation strategies.

Author Response

Authors' responses are attached.

Reviewer 2 Report

Comments and Suggestions for Authors

This study investigates the impact of environmental chemical pollutants on microbiome diversity in sediment and water samples from the Kinnickinnic, Milwaukee, and Menomonee Rivers near Milwaukee, Wisconsin. Using shotgun metagenomic sequencing and chemical analysis, the authors reveal that microbial density decreases with increasing chemical concentrations, while certain bacterial and archaeal populations show positive correlations with pollutants. The study highlights the potential roles of multi-kingdom microbial consortia in bioremediation and pollutant degradation. The research provides valuable insights into the relationship between chemical pollutants and microbial communities, particularly the differential responses of bacteria, archaea, fungi, and protozoa to pollution. This has significant implications for understanding microbial adaptation to contaminated environments and developing bioremediation strategies. However, after careful review, I recommend that the manuscript undergo revisions before acceptance. 

   - The study does not include control groups or reference sites with low or minimal pollution levels. Without such comparisons, it is challenging to definitively attribute the observed changes in microbial diversity and parameters (including viruses) to the presence of chemical pollutants, especially in a multi-pollutant context. How can the authors ensure that the observed correlations are not influenced by other environmental variables (e.g., temperature, pH, nutrient levels)? I strongly recommend incorporating control sites or historical data from less polluted areas to strengthen the conclusions.

   - The study presents a "snapshot" of microbial communities and pollutant levels at the time of sampling. However, water pollutants and microbial compositions are highly dynamic, and even sediment communities, though more stable, can vary over time. The lack of temporal analysis or historical data limits the ability to draw conclusions about long-term trends or cause-effect relationships. I suggest including historical data or conducting time-series sampling to provide a more comprehensive understanding of how microbial communities respond to pollutants over different time scales. Comparative analyses over time would significantly enhance the study's impact.

   - The quality of Figures 1 and 3 is notably poor, making it difficult to interpret the data presented. In contrast, Figure 5 is clear and well-presented. I recommend improving the resolution and clarity of Figures 1 and 3 to meet the journal's standards.

Author Response

Authors' responses attached.

Round 2

Reviewer 2 Report

Comments and Suggestions for Authors

Accept in present form